# Polymeric Microparticles: Synthesis, Characterization and In Vitro Evaluation for Pulmonary Delivery of Rifampicin

**DOI:** 10.3390/polym14122491

**Published:** 2022-06-19

**Authors:** Faiqa Falak Naz, Kifayat Ullah Shah, Zahid Rasul Niazi, Mansoor Zaman, Vuanghao Lim, Mulham Alfatama

**Affiliations:** 1Faculty of Pharmacy, Gomal University, Dera Ismail Khan 29050, Pakistan; faiqafalaknaz@gmail.com (F.F.N.); kifayatrph@gmail.com (K.U.S.); zahidscholar1@gmail.com (Z.R.N.); mansoor_zaman861@yahoo.com (M.Z.); 2Advanced Medical and Dental Institute, Universiti Sains Malaysia, Bertam, Kepala Batas 13200, Penang, Malaysia; 3Faculty of Pharmacy, Universiti Sultan Zainal Abidin, Besut Campus, Besut 22200, Terengganu, Malaysia

**Keywords:** aloe vera, sodium alginate, microparticles, rifampicin, dry powder inhaler

## Abstract

Rifampicin, a potent broad-spectrum antibiotic, remains the backbone of anti-tubercular therapy. However, it can cause severe hepatotoxicity when given orally. To overcome the limitations of the current oral therapy, this study designed inhalable spray-dried, rifampicin-loaded microparticles using aloe vera powder as an immune modulator, with varying concentrations of alginate and L-leucine. The microparticles were assessed for their physicochemical properties, in vitro drug release and aerodynamic behavior. The spray-dried powders were 2 to 4 µm in size with a percentage yield of 45 to 65%. The particles were nearly spherical with the tendency of agglomeration as depicted from Carr’s index (37 to 65) and Hausner’s ratios (>1.50). The drug content ranged from 0.24 to 0.39 mg/mg, with an association efficiency of 39.28 to 96.15%. The dissolution data depicts that the in vitro release of rifampicin from microparticles was significantly retarded with a higher L-leucine concentration in comparison to those formulations containing a higher sodium alginate concentration due to its hydrophobic nature. The aerodynamic data depicts that 60 to 70% of the aerosol mass was emitted from an inhaler with MMAD values of 1.44 to 1.60 µm and FPF of 43.22 to 55.70%. The higher FPF values with retarded in vitro release could allow sufficient time for the phagocytosis of synthesized microparticles by alveolar macrophages, thereby leading to the eradication of *M. tuberculosis* from these cells.

## 1. Introduction

Tuberculosis (TB) is currently one of the major health problems due to its higher rate of morbidity and mortality throughout the world [1]. It is an airborne communicable disease that occurs when healthy individuals inhale *Mycobacterium tuberculosis (M. tuberculosis*)-infected droplets exhaled into the air during the coughing and sneezing of a TB patient. These droplets are in the range of 2 to 3 µm, being suitable for deep lung inhalation [2,3]. Once Mycobacterium reaches the lung, alveolar macrophages engulf them by the processes of phagocytosis [4,5]. When *M. tuberculosis* is taken up by alveolar macrophages, it impairs phagosome maturation and hijacks the cellular processes to induce a necrotic, nutrient-rich environment that promotes its replication, thereby resulting in the formation of granuloma [6,7]. The disease mostly affects the lung and remains localized in 75 to 80% of cases but can also disseminate to other sites causing extrapulmonary tuberculosis [7,8]. The current treatment of TB is complicated due to oral and/or parenteral administration of multiple drugs for a prolonged period, which results in systemic toxicity and patient non-compliance problems.

Rifampicin is one of the most potent and broad-spectrum antibiotics against bacterial pathogens and is a key component of anti-tubercular therapy. Isoniazid and pyrazinamide have maximum efficacy in the extracellular and intracellular environment of the macrophages, respectively. Rifampicin is equally effective in both the extracellular and intracellular environment of the lung [9,10]. It is classified as a BCS class-II drug, having low solubility and high permeability [11]. Due to its lower solubility (solubility 2.50 mg/mL, log p 1.09, pKa 4.96 ± 0.70), rifampicin shows poor bioavailability in fixed as well as single-dose formulations [12]. Similarly, co-administration with other drugs, such as isoniazid, further reduces its bioavailability. Mere rifampicin administration as the inhaled formulation is expected not only to reduce its degradation and improve its bioavailability but will also lead to dose reduction and reduced toxicity due to direct deposition at the site of action.

The oral route of drug delivery remains dominant, but other routes of drug administration such as inhalation are becoming more popular for targeted drug delivery in the treatment of local respiratory and systemic diseases [13,14]. The advantages of pulmonary drug delivery concerning other routes of drug administration include the rapid onset of action, high drug concentrations in the lungs, lower enzymatic activity, avoidance of hepatic first-pass metabolism, lower systemic bioavailability and lesser systemic side effects [15,16,17]. It is a non-invasive route of drug administration, thereby avoiding patient compliance problems [18,19]. Pulmonary drug delivery has been used for many years for the treatment of lung diseases, and it has the potential to improve the treatment of TB [20,21]. The use of particulate carriers to be delivered via DPI is an attractive way of designing pulmonary drug delivery systems. These carriers can control drug release, ensure selective drug targeting to the intended site in the lung and offer improved interaction with biomolecules on both the cell surfaces and within the cells because of their comparable size to biological entities [22]. The behavior of inhaled particles and their deposition in the lung depends on several factors including particle dimensions, density, shape, composition, concentration, surface properties such as particle charge, and the breathing pattern of the individual [23,24].

Currently, the carriers for therapeutic molecules are mostly fabricated using natural polymers due to their abundance, versatility and lower probability of immunogenic responses [25]. Sodium alginate has been extensively investigated for pulmonary drug delivery due to its muco- and bio-adhesive properties, biocompatible and biodegradable nature and nontoxic characteristics [26,27]. It is thermally stable and can be used for the encapsulation of substances with the necessity of prolonged and controlled release [28,29]. However, sodium alginate had been previously found to possess several limitations of poor mechanical properties and microbial degradation due to its hydrophilic nature [30,31]. To overcome these problems, aloe vera and L-leucine were also used in the fabrication of spray-dried powder formulations. The anti-tubercular activity of aloe vera has been well documented [32] and was previously used as adjuvant therapy to improve the efficacy of conventional anti-mycobacterial therapies, decrease their adverse effects due to hepatoprotective properties and reverse the multidrug resistance. The essential amino acid L-leucine has been widely used in the formulation of dry powder and is generally regarded as a safe excipient for pulmonary administration [33]. It increases the aerosolization of drugs from DPI by decreasing surface free energy and improving the physical stability of DPI [14,34]. L-leucine potentially slows the drug release in the lungs due to its hydrophobic nature [35] and improves the percentage yield of spray-dried particles [36,37].

Spray drying allows the synthesis of microparticles with the desired physicochemical properties by optimizing processing parameters [38]. Spray drying is a simple, fast and one-step processing of microparticles synthesis for pulmonary delivery [39]. It is a reproducible, affordable, time saving, continuous and scalable process [40,41]. The long shelf-life stability is largely attributed to the low moisture content of the spray-dried powders [42]. Spray-drying produces a fine powder of the desired size, shape, morphology and surface texture for better aerosolization performance and allows for the large-scale production of powders with high batch-to-batch reproducibility [43,44]. Thus, the current study focuses on designing spray-dried polymeric microparticles of the desired physicochemical properties that can achieve optimum aerodynamic characteristics concerning aerosolization and inhalation.

## 2. Materials and Methods

### 2.1. Materials

Rifampicin (Sigma Aldrich, Rockville, MD, USA) was used as a drug of choice, sodium alginate (Sigma Aldrich, Rockville, MD, USA) and aloe vera powder (MINATURE, MARUDHRA IMPEX, Ahmadabad, India) were utilized as matrix former, L-leucine (Sigma Aldrich, Rockville, MD, USA) was added as spray drying excipients. Phosphate buffer saline (Sigma Aldrich, Rockville, MD, USA) was used as a dissolution medium to mimic lung fluid. Methanol (Sigma Aldrich, Rockville, MD, USA) was used as a solubilizing medium for rifampicin. All other chemicals used were of analytical grade and utilized without any further purification.

### 2.2. Methods

The preparation of microparticles with the desired physicochemical properties was conducted using a Pilotech YC-015 spray-dryer (Pilotech Equipment Co., Ltd., Shanghai, China) [45]. The obtained particles were characterized for their suitability for inhalation drug delivery concerning size and size distribution, morphology, roughness, circularity, density, flow characteristics, drug association efficiency and release behavior. Finally, the powder was evaluated for its aerosolization and inhalation performance using Anderson cascade impactor (ACI).

#### 2.2.1. Optimization of Formulation and Spray-Drying Conditions

The development of drug carriers for inhaled medicine is challenging due to the required suitable physicochemical properties for deposition in the desired region of the lungs [46]. In the case of tuberculosis, the mycobacterium tuberculosis invades alveolar macrophages in the peripheral lung and multiplies inside these cells [38,47]. To target the alveolar macrophages harboring mycobacteria, particles of aerodynamic diameter in the range of 1 to 5 µm and geometric diameter of 1–3 µm were aimed to be synthesized. To fabricate particles of the desired characteristics, the formulations were optimized using variable concentrations of sodium alginate and L-leucine at constant rifampicin and aloe vera powder concentrations. In the optimized batch of the first five formulations, the concentration of L-leucine was gradually increased from F1 to F5 in a ratio of 1:1; 1:2; 1:3; 1:4 and 1:5, respectively. In the next batch of experiments, the concentration of sodium alginate was gradually decreased (1:5; 1:4; 1:3; 1:2 and 1:1) from F6 to F9 as mentioned in Table 1. The spray drying allows the engineering of microparticles with the desired physicochemical properties by optimizing processing parameters such as feed rate, atomizing air pressure, inlet and outlet air temperature, etc. [38]. The spray drying conditions were optimized in initial experiments and used as follows: inlet air temperature 120 °C, outlet temperature = 70 to 90 °C, solution feed rate = 2 mL/min and atomizing air pressure = 5.5 bar.

#### 2.2.2. Synthesis of Microparticles

The microparticles were prepared using variable concentrations of sodium alginate and L-leucine at the constant therapeutic dose of rifampicin and aloe vera powder concentration. Briefly, the rifampicin was dissolved in methanol at 25 ± 1 °C under continuous stirring at 1000 rpm for 1 h. The L-leucine and aloe vera powder were dissolved in distilled water until a clear solution was obtained. Sodium alginate solution was separately prepared in distilled water and added dropwise to L-leucine and aloe vera powder solution at constant magnetic stirring for 1 h. Finally, the rifampicin solution was added dropwise to form the final mixture of spray drying feed solution. The feed solution was pumped to a stainless-steel inner nozzle (tip diameter 0.7 mm) at optimized spray-drying conditions as mentioned previously. The spray-dried powders were retrieved using a rubber spatula into a 10 mL amber diagnostic vial and kept in a desiccator at room temperature (25 ± 1 °C) till further use. The percentage yield was calculated concerning the total solid content of the feed solution as mentioned in Equation (1). It depicts the efficiency of the spray dryer in producing DPI formulation for inhalation drug delivery.
(1)$$\mathrm{Percentage}\;\mathrm{yield}=\frac{\mathrm{Total}\;\mathrm{weight}\;\mathrm{of}\;\mathrm{powder}\;\mathrm{obtained}}{\mathrm{Total}\;\mathrm{solid}\;\mathrm{content}\;\mathrm{of}\;\mathrm{feed}\;\mathrm{solution}}\times 100$$

#### 2.2.3. Size and Size Distribution

The particle size and size distribution of the spray-dried particles were evaluated using a Horiba particle size analyzer (Horiba Ltd., LA-300, Kyoto, Japan). The sample was suspended in HPLC grade cyclohexane, added to the liquid dispersion unit of the Horiba particle size analyzer to determine the particle size distribution by measuring the intensity of light scattered as a laser beam passes through a dispersed particulate sample. The median particle sized (0.5) was reported based on volume distribution.

#### 2.2.4. Zeta Potential

The zeta potential of the particles was determined using Zetasizer (Malvern Zetasizer Nano ZS 90, Malvern Instruments Ltd., Malvern, UK). The sample was dispersed in phosphate buffer (pH 7.4), and an aliquot of 700 µL was introduced to the zeta potential cell for analysis of surface charge of spray-dried particles.

#### 2.2.5. Morphology

The morphology of the spray-dried particles was evaluated using a scanning electron microscope (SEM). Briefly, 2 mg of sample was spread on double-sided carbon adhesive taps attached to aluminum stubs. The excess particles were removed by gentle tapping, followed by platinum coating at a current intensity of 20 mA using an auto fine coater (JFC1600, JEOL, Tokyo, Japan). The representative sections were photographed at various magnifications.

#### 2.2.6. Roughness and Circularity

The roughness and circularity of particles were determined using processing software ImageJ (NIH, Bethesda, MD, USA) [48]. Briefly, the SEM photograph of each formulation was selected at suitable magnification (2700×) and processed for calculation of roughness and circularity values quantitatively using the specific plugin. The values were shown as an average of five readings.

#### 2.2.7. Powder Density and Flow Properties

The flow property of powder formulation is a direct indicator of dispersed fraction from DPI device and was determined from the bulk and tapped density of powder as mentioned previously [49]. Briefly, the weight of the spray-dried powder was determined using a microbalance (Metler Tolendo Inc., model-MX5, Urdorf, Switzerland). The microparticles of known weight were poured under gravity in a 5 cm^3^ graduated cylinder. The sample in the graduated cylinder was gently tapped until no measurable change in the occupied volume of powder was noticed. The bulk and tapped density of the powder were expressed as the quotient of the weight (g) to the bulk and tapped volume, respectively. Carr’s index and Hausner’s ratio were calculated using Equations (2) and (3), respectively.
(2)$$\mathrm{Carr’s}\;\mathrm{index}=\frac{\mathrm{Tapped}\;\mathrm{density}-\mathrm{Bulk}\;\mathrm{density}}{\mathrm{Tapped}\;\mathrm{density}}\times 100$$
(3)$$\mathrm{Hausner’s}\;\mathrm{ratio}=\frac{\mathrm{Tapped}\;\mathrm{density}}{\mathrm{Bulk}\;\mathrm{density}}$$

All the readings were taken as a mean of at least three readings.

#### 2.2.8. Drug Content and Association Efficiency

An accurately weighed 5.0 mg of microparticles were dissolved in 100 mL of phosphate buffer (pH 7.4). The solution was sonicated three times for 30 s with a gap of 2 min between each sonication cycle to ensure the complete release of the drug from the spray-dried powders. The resultant solution was filtered through a syringe filter (pore size = 0.45 µm; Durapore, Millipore corporation, Cork, Ireland). The samples were analyzed spectrophotometrically at λmax of 333 nm, and rifampicin content was determined from the absorbance recorded concerning similarly blank microparticles solution as a control. The drug content was expressed as the quantity of drug encapsulated in a unit weight of spray-dried microparticles. The drug association efficiency was calculated as a percentage of rifampicin associated with spray-dried microparticles concerning the amount of rifampicin used in the formulation. The results were computed as the mean of triplicate analysis for each batch of microparticles.

#### 2.2.9. Crystallinity of Powder

The interaction of the drug with the polymer and physical nature of spray-dried rifampicin and fabricated microparticles were analyzed using an X-ray powder diffractometer (JDX-3532, JEOL, Tokyo, Japan). A radiation source of Cu-Kα (Wavelength = 1.5418 A°) at 40 kV voltage and 30 mA current was employed. The sample was placed in an aluminum sample holder and scanned over the range between 4 to 80° and position 2 Theta (ɵ) with increments of 0.008° at a rate of 3°/min at an ambient temperature. The raw data is converted into the graphical form using Origin Pro^®^ 2021 software (OriginLab Corporation, Northampton, MA, USA) [50]. The percentage crystallinity index (% CI) was determined using Equation (4). Mean ± SD was calculated from three measurements.
CI (%) = (I 110 − I amour) × 100/I110(4)
where I110 is the maximum intensity at 20° and I amour is the intensity of amorphous diffraction at 16° [48].

#### 2.2.10. Thermal Analysis

The thermal properties of spray-dried rifampicin and spray-dried microparticles were evaluated by differential scanning calorimetry (DSC 250, TA Instruments Thermal Analysis-DSC-TGA Standard, New Castle, PA, USA). Approximately 3 mg of powder was weighed in standard heating aluminum pans (Tzero low mass), crimped with Tzero lids (TA instrument crimper) and exposed to heating ramping at 30 °C/min. The samples were scanned on a heating range of 25 to 600 °C followed by a cooling cycle from 600 to 25 °C at the same heating rate (30 °C/min) under nitrogen gas, at a flow rate of 20 mL/min in the presence of empty aluminum pan as reference. Thermograms were integrated using software supplied by the manufacturer (Program Universal V4. 5A, cell constant 1.0060). The images at every unit degree change in temperature were recorded during the analysis. The raw data was then analyzed in graphical form using Origin Pro^®^ 2021 software (OriginLab Corporation, Northampton, MA, USA) [51].

#### 2.2.11. FTIR Analysis

The FTIR spectra were noted for raw rifampicin, aloe vera powder, sodium alginate, L-leucine and spray-dried microparticles (Perkin Elmer, FTIR spectrometer, Spectrum TWO LITA, Llantrisant, UK). The powdered samples (approximately 2 mg) were processed by the potassium bromide (KBr) pellet method in triplicate over the wavelength of 450 to 4000 cm^−1^, using a spectral resolution of 1 cm^−1^. An average of 16 scans per sample was recorded. Spectra were plotted and analyzed using Origin Pro^®^ 2021 software (OriginLab Corporation, Northampton, MA, USA) [50].

#### 2.2.12. The In Vitro Drug Release Profile

The in vitro release of rifampicin from spray-dried particles was studied using the dialysis membrane method [52]. The microparticles (containing 50 mg rifampicin) were introduced into activated dialysis membrane (MWCO 12000) containing 3 mL of phosphate buffer as release media (inner dissolution medium). The dialysis bag was sealed and placed in a larger vessel containing 300 mL of phosphate buffer (pH 7.4) as dissolution media mimicking the physiological conditions of the lung (outer media/compartment). The microparticles-filled dialysis bag was agitated at 50 rpm in the dissolution media in a thermostatically controlled shaking water bath at temperature 37 ± 2°. An aliquot of 5 mL was withdrawn at specific time intervals of 0.5, 1, 2, 4, 8, 16 and 24 h. The samples were filtered using syringe filters and analyzed spectrophotometrically at λ-max 333 nm using similarly processed blank microparticles as control. The samples were stirred overnight and exposed to sonication at least three times for 30 s at a gap of 2 min to completely extract the drug from the microparticles. The cumulative percentage of drug release was then plotted against the time of drug release. The in vitro release data were fitted in various kinetic release models to calculate the values of r^2^ and K to determine the relevant drug release mechanism for the linear curve obtained from regression analysis.

#### 2.2.13. Aerodynamic Behavior

The aerodynamic behavior of microparticles was determined using an Anderson cascade impactor (Copley Scientific, Nottingham, UK) equipped with an induction port and pre-separator that fractionates the particles according to their aerodynamic diameter. Briefly, 20 mg of microparticles were added to the Size “2” hard gelatin capsule (Gelcap Pakistan LTD, Karachi, Pakistan). The filled capsule was introduced into the sample chamber of Handihaler^®^ (Boehringer Ingelheim, Germany), pierced to activate the powder for inhalation into ACI. The airflow rate was modulated in such a way to produce a pressure drop of 4 kPa over the Handihaler, for the duration (T = 240/Qout, in seconds) consistent with the withdrawal of 4 L of air from the mouthpiece of the inhaler within the ACI. The capsule content was discharged from Handihaler^®^ by piercing the hard gelatin capsule, the amount of drug deposited in each stage of ACI was rinsed with phosphate buffer (pH 7.4), dissolved properly, filtered and analyzed for rifampicin content following analytical protocol as mentioned for drug content analysis. The data was then processed to determine matrices that include mass median aerodynamic diameter (MMAD), geometric standard deviation (GSD), fine particle fraction (FPF) and a respirable fraction (RF).

#### 2.2.14. Statistical Analysis

All the results were presented as mean ± SD for triplicate experiments. Student *t*-test was used to compare the results of two formulations, while the effect of each parameter on aerosolization and inhalation performance was determined using ANOVA (SPSS version 2.1).

## 3. Results and Discussion

### 3.1. Synthesis of Microparticles

The rifampicin-loaded microparticles were prepared using aloe vera powder as an immune modulator to circumvent the side effects of rifampicin with varying concentrations of alginate and L-leucine by the spray-drying method. In the nine designed formulations, five formulations (F1–F5) were fabricated with increasing concentrations of L-leucine, while the remaining four formulations contained increasing concentrations of sodium alginate (F9 to F6; Table 1). The addition of L-leucine (5 to 40% *w*/*w*) is known to control the inter-particulate interactions and improve the aerosolization of small particles without the need for a coarse carrier [33]. The spray drying technique is commonly used in the preparation of polymeric microparticles due to reliability, reproducibility and control of particle size, shape and drug release characteristics [53]. It is a one-step continuous process that can be applied to both hydrophilic and hydrophobic polymers and/or drugs, is easy to scale up and can be applied for both heat-resistant and heat-sensitive drugs [54]. The percentage yield of spray-dried powder was in the range of 45 to 65% (Table 2). Generally, the percentage yield of spray-dried powder decreased from F1 to F3 due to the surfactant nature of L-leucine that decreases the surface tension thereby leading to the atomization of small particles exposed to inlet air temperature of 120 °C. The small droplets thus dried and deposited as an agglomerated mass in the collection bottle (Figure 1; F1–F4). When the concentration of L-leucine in formulations (F5) was increased, the percentage yield was also improved due to the hydrophobic coating of L-leucine on the surface of spray-dried particles that were deposited as individual particles. The results of our study were in coherence with the previous study, whereby L-leucine incorporated formulations achieved a higher yield of spray-dried microparticles [36]. The higher yield of spray-dried microparticles as a function of alginate concentration could be contributed to a corresponding increase in feed solution viscosity, which has been shown to affect particle size (Table 2) [55]. The higher density of spray-dried microparticles could also contribute to a higher yield of spray-dried microparticles (F9–F6).

### 3.2. Particle Size and Zeta Potential

The size and size distribution of microparticles prepared by spray-drying feed solution containing different concentrations of L-leucine (F1–F5) and sodium alginate (F6–F9) is presented in Table 2. The particles size of the initial three formulations increased with as the concentration of L-leucine increased in spray drying feed solution. This was due to the higher solubility of L-leucine in the feed solution, thereby drying within the matrix of spray-dried powder (F1–F3; Table 2). The resultant particles were more cohesive with high tendency to aggregate as shown from particle size distribution data (Table 2). When the concentration of L-leucine was further increased, (F4–F5; Table 2) it migrated to the surface of the drying droplet [56] and the resultant spray-dried microparticles were less aggregated with reduced particle size distribution. L-leucine has been previously shown to achieve substantially higher concentrations on the surface than the bulk of spray-dried microparticles due to a significant reduction in surface energy of spray-dried formulations [57]. The formulation with a higher L-leucine concentration (F5) enhanced the yield of the spray-dried microparticles [36]. The formulations (F6–F9) containing a higher amount of sodium alginate exhibited increased particle size and lower particle size distribution due to effective encapsulation of the drug within the highly viscous spray-drying feed solution. The particle size of all synthesized formulations ranged from 2.0 to 4.0 µm, being optimum for deep lung inhalation and passive drug targeting to alveolar macrophages [58]. The literature reports preferential macrophage uptake of particles with sizes between 1 and 3 µm [47]. The zeta potential of spray-dried particles depicts that most of these particles were negatively charged due to the presence of sodium alginate in the formulations. However, pure drug microparticles and F5 showed positive zeta potential due to charge masking properties of non-polar L-leucine at higher concentrations (Table 1 and Table 2).

### 3.3. Morphology, Roughness and Circularity

The surface morphology, roughness and circularity of microparticles can affect the aerosol dispersion from the DPI device as well as the deposition of particles in the lung. The shape of synthesized particles is an important feature in the macrophage uptake process. It was reported that spherical particles are readily internalized by macrophages, whereas elongated particles may hinder phagocytosis [18,59]. The surface morphology of microparticles prepared by spray drying feed solution of either pure drug or drug containing polymer blends is presented in Figure 1. The spray-dried pure rifampicin microparticles were mostly elongated with cylindrical shape. The circularity of particles increased while agglomeration tendency of the powder decreased with increasing concentration of L-leucine in formulation F1–F3 (Table 2). The first four formulations had irregular morphology with wrinkles on their surfaces. In the case of F5, where L-leucine concentration is maximized with respect to sodium alginate (1:5), completely flocculated round shape monodisperse particles were obtained (Figure 1). In F9, having sodium alginate to L-leucine ratio of 2:1, the obtained particles were completely spherical with smooth surfaces. The spherical shape of the microparticles formed as the droplets dry in the stream of hot air, is a characteristic of amorphous powder [54]. As the concentration of sodium alginate increased with respect to L-leucine in spray drying feed solution (F8 to F6), the particles became porous inside and rounded with cup-shaped morphology (Figure 1). The roughness of spray-dried pure drug microparticles was significantly higher than drug encapsulated polymeric drug microparticles (*p* ≤ 0.05; Table 2). The rough surfaces at nanoscale in the case of F5, F6 and the pure drug could reduce the cohesion/adhesion forces between approaching surfaces of individual particles, thereby playing a role in improved aerosolization and inhalation performance of spray-dried microparticles [60]. Thus, the addition of L-leucine and/or sodium alginate not only affects the size, but also the surface properties of spray-dried particles as mentioned previously [53].

### 3.4. Powder Density and Flow Properties

The flow properties of powder formulations depend on numerous factors that include particle size distribution, the shape of the particle, chemical composition, moisture content and temperature. The bulk density, tapped density, compressibility index and Hausner’s ratio are primary parameters considered to evaluate the flow of powder from dry powder inhalers [61]. The free-flowing powder has bulk and tapped density values closer to each other, thereby giving Carr’s index value of less than 15, while cohesive powder would have higher Carr’s index values due to significant variation in bulk and tapped density [62]. The tapped density of formulations (F9-F6) containing higher concentrations of sodium alginate with respect to L-leucine was lower than other formulations (F1–F5) due to the formation of hollow particles. The formulations whereby L-leucine (F5) or sodium alginate (F8–F6) concentration was maximized produced a powder that can achieve comparatively higher flowability as depicted from their lower Carr’s index and Hausner’s ratio (Table 3). This was due to the presence of small particles that can act as a glidant between the powder formulations of DPI. In the case of pure spray-dried rifampicin microparticles, the particles were cylindrical in shape that lowers the overall contact between individual particles. Carr’s index values in the range of 37 to 65 and Hausner’s ratio higher than 1.50 depicted the cohesive nature of spray-dried particles due to their small particle size.

### 3.5. Drug Content and Association Efficiency

The drug content and drug association efficiency of the spray-dried rifampicin microparticles are presented in Table 3. The drug content of microparticles ranged from 0.24 to 0.39 mg/mg of powder, with drug association efficiency of 39.28 to 96.15%. The drug association efficiency of spray-dried particles increased significantly with increasing L-leucine concentration (F1–F5) as well as sodium alginate (F9–F6). Rifampicin, being a class-II drug, mainly remains entrapped in the polymer matrix at a higher polymer concentration that protects the drug from degradation at a higher inlet air temperature.

The formulations (F4, F5, F6 and F7) having maximum drug association efficiency, optimum particle size and comparatively better flow properties were further evaluated for their drug release and aerodynamic properties using ACI.

### 3.6. XRD Analysis

The crystalline and/or amorphous nature of spray-dried powders of pure rifampicin and formulations containing a higher concentration of L-leucine (F4 and F5) or sodium alginate (F6 and F7) were evaluated using X-ray diffractogram as shown in Figure 2. The diffractogram was plotted against position 2 theta and intensity 4 to 80 on an arbitrary scale. When individual formulation was evaluated, L-leucine was found to be crystalline, while rifampicin, sodium alginate and aloe vera powder were amorphous in nature. The XRD diffractogram confirmed that all the formulations were amorphous in nature with some crystalline peaks of 5.59° and, 5.6°, 18.90° and 24.02°, respectively, in F4 and F5. The F4 and F5 formulations have crystalline peaks because they contain the highest concentration of L-leucine. The primary peak 6° (5.59, 5.6 and 5.91, 5.62) is apparent in all the formulations (Figure 2). There were no sharp peaks in F6 and F7, having a higher concentration of sodium alginate with respect to L-leucine. Thus, L-leucine overall contributes to the crystallinity of powder in spray-dried formulations. The influence of L-leucine on improving dispersibility and aerosolization properties has been previously revealed [63,64]. When the crystallinity of formulations was compared to pure spray-dried rifampicin (cCI = 25%), the formulation was found be to more crystalline (cCI = 27–36%). Theoretically, the amorphous form exists in a higher free energy state and is thus a metastable form with higher water solubility compared to the crystalline form, which is more stable [65]. Overall, the XRD analysis showed successful incorporation of the drug into the polymeric matrix was achieved, which further improves the stability of rifampicin.

### 3.7. DSC Analysis

The DSC thermogram of pure spray-dried rifampicin and rifampicin-loaded polymeric microparticles (Figure 3) was conducted to evaluate the effect of sodium alginate and L-leucine on thermal properties of the drug. Rifampicin existed in two polymorphic forms (form I and II), where form-I has lower chemical stability and quickly decomposes before melting at 140–160 °C. The meta-stable form-II rifampicin melted at 189 °C and re-crystallized as a form-I at 204 °C [66]. In the DSC thermogram, a broad endothermic peak can be seen in the range of 140 to 160 °C, indicating the melting of spray-dried amorphous rifampicin (Figure 3 (a)). The pattern of melting endotherm was observed for L-leucine-containing formulations F4 and F5 (Figure 3 (b) and (c)). In the case of formulations with higher sodium alginate concentration (F6 and F7), the melting peak was more broadened due to the higher concentration of entrapped drugs in the polymeric matrix. The sharp exothermic peak observed at 225 °C in pure spray-dried rifampicin and formulations with higher L-leucine concentration (F4 and F5, Figure 3 (a), (b) and (c)) signified the re-crystallization of rifampicin as form-I. The subsequent peaks observed at 250 °C and above represents the decomposition of rifampicin [65]. The lack of sharp peaks or presence of shallow peaks in all the samples revealed the amorphous nature of the powders which is in line with the XRD diffractogram.

### 3.8. FTIR Analysis

The ATR-FTIR spectra of raw materials and fabricated microparticles (F4–F7) are shown in Figure 4. The presence of characteristic peaks in each of the raw material depicts the purity of material used in the fabrication of microparticles. The characteristic bands of rifampicin appeared at 3477 cm^−1^ (-OH), 1728 cm^−1^ (furanone), 1644 cm^−1^ (amide near C-O), 1567 cm^−1^ (C=C) and 1490 cm^−1^ (amide close to the C-C). The aloe vera powders have shown spectral bands at 3350 cm^−1^, that attributed to the stretching vibration of the hydroxyl group, and the peak that appeared at 1599 cm^−1^ was attributed to the amino groups present in alcohol, phenol and amines [67,68,69]. In the case of sodium alginate, the functional group region of 3480 cm^−1^ and 2920 cm^−1^ were related to the OH stretching mode of the hydroxyl group and the CH stretching, respectively [70]. The characteristic peak of L-leucine at 2953 cm^−1^ was due to aliphatic CH_3_ bending, a peak at 1577 cm^−1^ could be allotted to the N-H band, while the band at 1399 cm^−1^ appeared due to COO−asymmetric vibration (Figure 4) [71,72]. In the rifampicin-loaded formulation with a higher concentration of L-leucine (F4 and F5), all the characteristic bands of the drug appeared that indicate the chemical stability of rifampicin even after encapsulation into the polymer matrix (Figure 4). However, the characteristic peak of L-leucine for aliphatic CH_3_ bending at 2953 cm^−1^ was more prominent due to a higher concentration of L-leucine (Figure 4). In the case of formulation with a higher concentration of sodium alginate (F6 and F7), the 3420 cm^−1^ (OH stretching) and 2918 cm^−1^ (CH stretching) bands appeared at higher intensity due to some level of chemical interaction between formulation components. Overall, the FTIR spectra confirmed the presence and integrity of raw materials during the spray drying process of microparticles synthesis, and no potential interaction and chemical incompatibility were found.

### 3.9. Drug Dissolution and Mechanism of Drug Release

The drug dissolution profile of rifampicin from selected spray-dried microparticles was shown in Figure 5. Overall, the release of the drug from microparticulate formulations was significantly controlled in comparison to pure spray-dried rifampicin which achieved complete dissolution within 8 h. The sustained release of the drug from polymeric particles (F4–F7) was attributed to the slow penetration of the dissolution medium through the polymeric coating around the drug. Rifampicin is a BCS class-II drug, and its dissolution is mainly dependent on formulation components. In the case of formulations (F4–F5) with higher L-leucine concentration, the release was significantly retarded when compared to formulations with higher concentrations of sodium alginate (*p* ≤ 0.05; Figure 5), due to the hydrophobic nature of L-leucine. The amorphous powder has a higher solubility and faster dissolution rate than crystalline; therefore, the crystalline nature of F5 could also be responsible for controlled drug release [73]. Overall, the slow dissolution of drug from spray-dried particles could allow sufficient time for phagocytosis of these particles by alveolar macrophages thereby leading to the eradication of *M. tuberculosis* from these cells. The in vitro release of rifampicin from spray-dried microparticles can be best explained by the Korsmeyer–Peppas model showing maximum linearity (0.9756) as mentioned in Table 4. The *n* value in the Korsmeyer–Peppas model varied between 0.60 and 0.85, suggesting a diffusion-controlled (non-Fickian) release [74,75].

### 3.10. In Vitro Aerodynamic Behavior

The matrices that depict the aerodynamic performance of rifampicin-loaded microparticles prepared from aloe vera powder with variable concentrations of sodium alginate and L-leucine are shown in Table 5. The emitted dose (ED) was found to be in the range of 60 to 70%, comparatively higher in the case of F4 than F5 due to large particle size (Table 5 vs. Table 2). The same pattern of higher dispersed fraction (PD) was observed when F6 was compared to F7. Previous studies have shown that any increase in particle size increases the aerosol dispersion from the DPI device [76]. However, complete dispersion of drug from DPI device could not be achieved due to the cohesive nature of microparticles having the propensity to aggregate, thereby showing poor aerosolization performance [77]. Both roughness and circularity of particles had been previously shown to affect the aerosolization of drug from DPI’s devices [76]. The roughness at nanoscale promotes higher dispersion of aerosol mass due to smaller aggregation strength between the approaching surfaces of microparticles [60]. However, our results were in contradiction to the previous finding due to the dominating effect of other physicochemical properties. The MMAD of designed particles are in close proximity to theoretical aerodynamic diameter (Table 5). The particles in the range of 1 to 2 µm had been shown to be best suited for deposition in the peripheral lung [78,79]. The MMAD of microparticles formulations (F4–F6) achieved in the range of 1.44 to 1.60 µm will promote particle deposition in the deep lung by the mechanism of sedimentation and diffusion [77,80]. When inhalation performance was evaluated using ACI, significantly higher inhaled fraction (PI) along with FPF and RF was achieved for F7, in comparison to other formulations due to higher circularity, lower tapped density and the presence of fine particles among large particles that promotes its deposition in the lower lung. The deposition pattern of spray-dried polymeric microparticles has shown that powder exists in the form of soft agglomerates due to cohesive nature that promotes < 25% aerosol mass to be deposited in induction port (throat) and pre-separator (Figure 6). The majority of aerosol mass gets deposited on stage 4 to stage 7, corresponding to aerosol mass with an aerodynamic diameter of 2.53 µm to 0.31 µm. When aerosol particles deposit in the alveolar region, alveolar macrophages get activated and engulf the deposited particles encapsulating rifampicin by means of phagocytosis [81,82]. Thus, the amount of drug in alveolar macrophages harboring *Mycobacterium tuberculosis* could be maximized by designing optimum polymeric microparticles to achieve effective eradication of *mycobacterium* bacilli from the lung.

## 4. Conclusions

In this study, inhalable rifampicin-loaded polymeric microparticles were prepared using the spray drying technique. The concentration of L-leucine was gradually increased in the first five formulations (F1–F5), while in the case of the last five formulations (F9-F6), the sodium alginate content was gradually increased, keeping all other formulations components at a constant level. It has been observed that F4, F5, F6 and F7 were lead formulations due to their desired physicochemical properties with respect to particle size (1.44 to 1.59 µm), rounded morphology (circularity ≥ 0.7), flow characteristics and drug content. The dissolution data depicts that F4 and F5 with a higher L-leucine concentration have significantly retarded the release of rifampicin from microparticles compared to formulations with a higher concentration of sodium alginate (F6 and F7), due to the hydrophobic nature of L-leucine. The amorphous powder has a higher solubility and faster dissolution rate than crystalline; therefore, the crystalline nature of F5 could also account for the controlled drug release. The “*n*” value in the Korsmeyer–Peppas model varied between 0.60 and 0.85, suggesting a diffusion-controlled release mechanism. The slow release of a drug from spray-dried microparticles could allow sufficient time for phagocytosis of these particles by alveolar macrophages, thereby leading to the eradication of *M. tuberculosis* from these cells. The aerosol performance of the spray-dried microparticles was assessed using ACI and found that microparticles achieved optimum MMAD of around 1.42 to 1.59 for deep lung deposition with comparatively higher aerodynamic particle size variation as depicted by GSD values of 3.0 to 4.0. The deposition of the majority of aerosol mass at stage 3 to stage 7 mimics deeper parts of the lung as shown by higher FPF and RF values which requires further studies to scale-up rifampicin powder formulations for inhalation drug delivery.

## Figures and Tables

**Figure 1 polymers-14-02491-f001:**
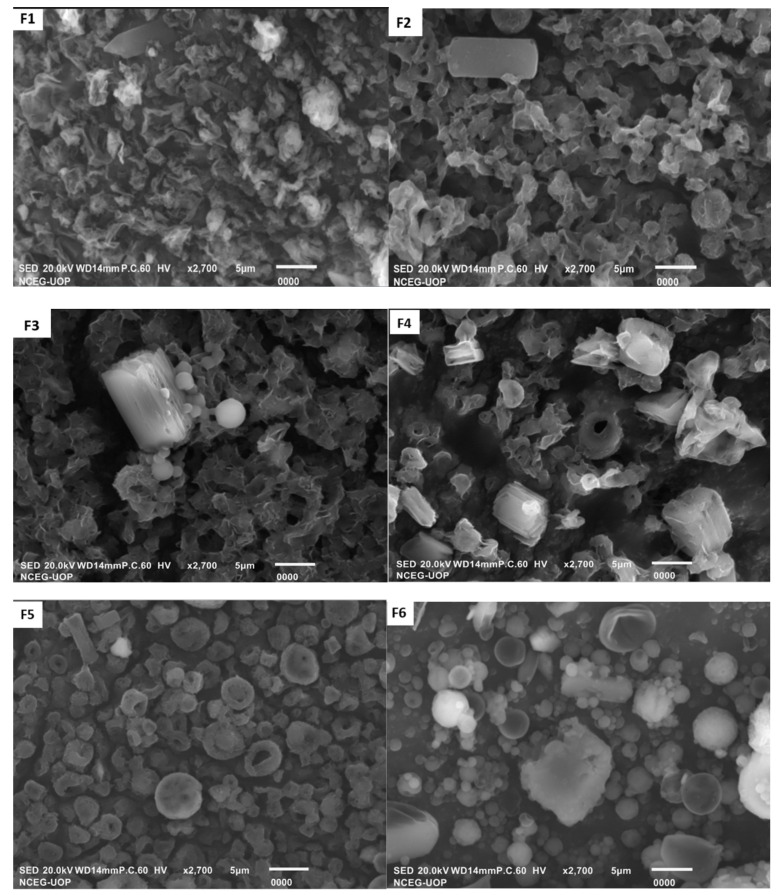
Morphological features of spray-dried microparticles (F1–F5) and pure rifampicin (R) and spray-dried, rifampicin-loaded microparticles (F6–F9).

**Figure 2 polymers-14-02491-f002:**
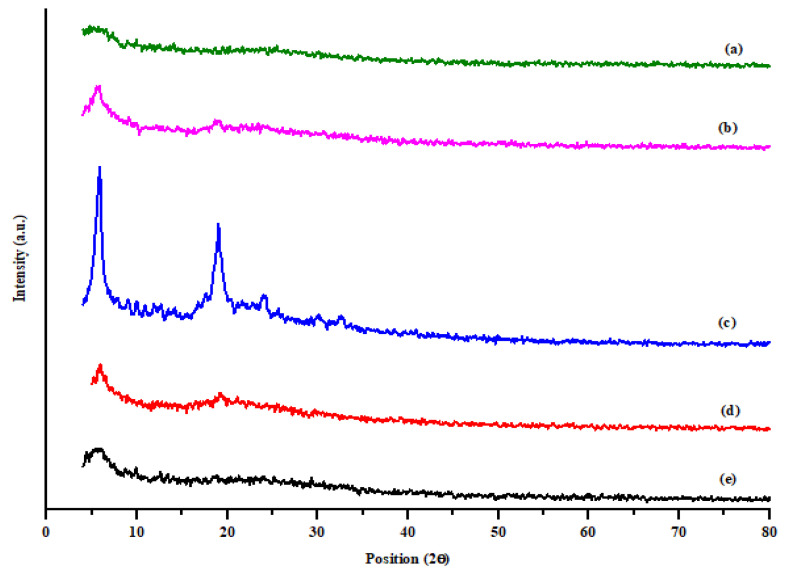
XRD diffractogram of (a) SD rifampicin (b) F4, (c) F5, (d) F6, (e) F7 at position 2 theta and intensity 4 to 80 (arbitrary scale).

**Figure 3 polymers-14-02491-f003:**
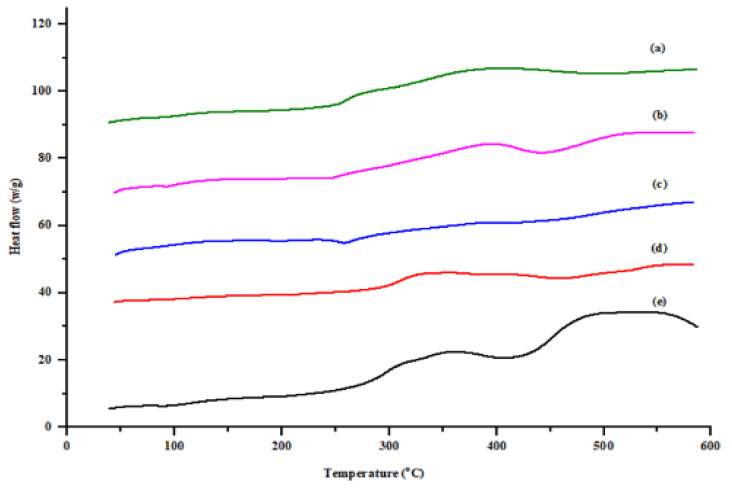
DSC thermogram of (a) spray-dried rifampicin (b) F4, (c) F5, (d) F6 and (e) F7.

**Figure 4 polymers-14-02491-f004:**
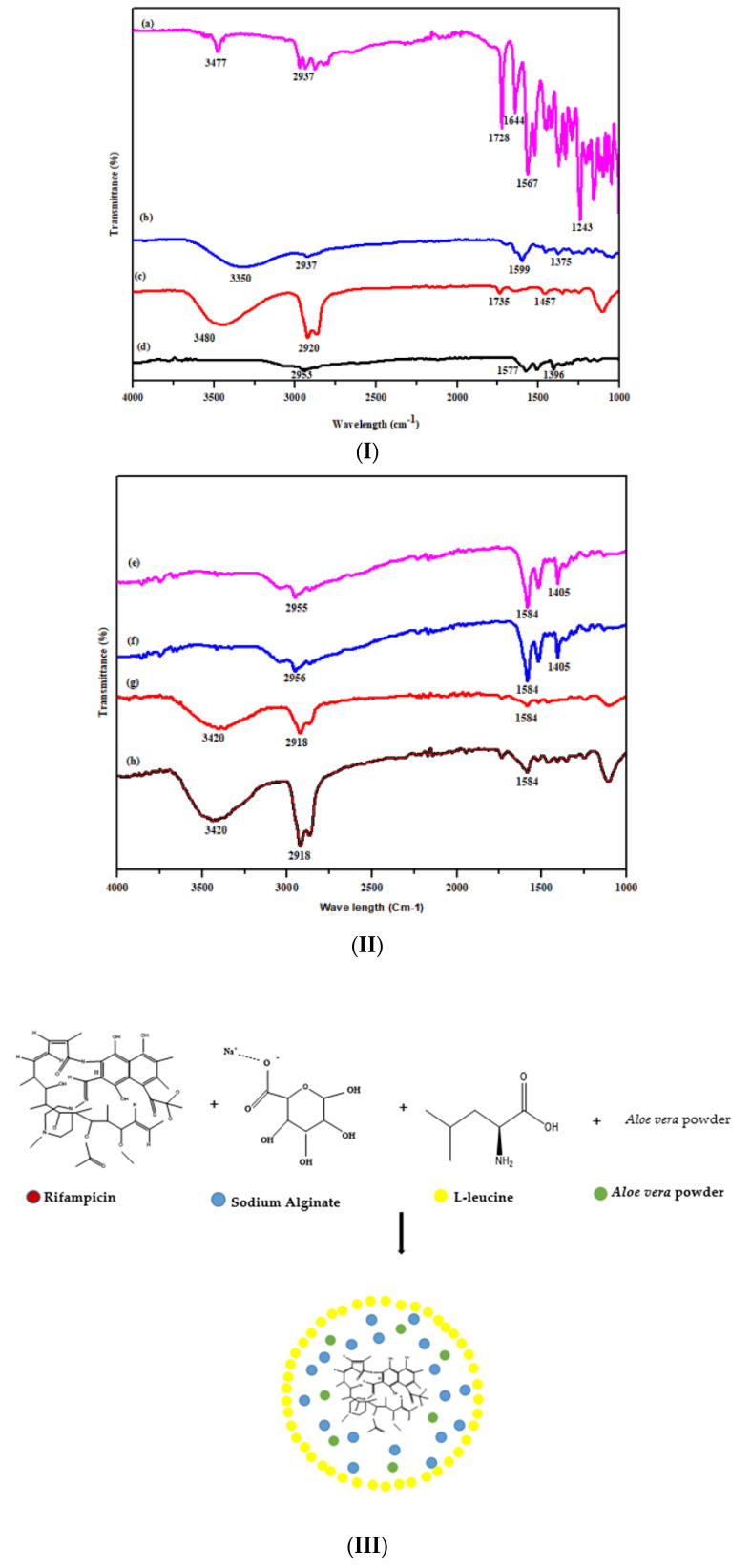
(**I**). FTIR Spectra of rifampicin (a), aloe vera (b), sodium alginate (c) and L-leucine; (**II**) (d) spray-dried microparticles formulations F4 (e), F5 (f), F6 (g) and F7 (h); (**III**) schematic representation of molecular structures of drug, carriers and spray-dried microparticles.

**Figure 5 polymers-14-02491-f005:**
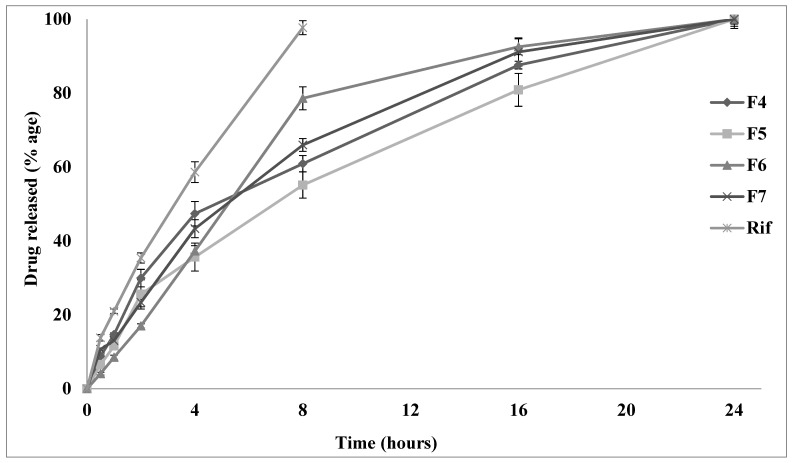
In vitro release profile of rifampicin from spray-dried microparticles (F4–F7) and pure rifampicin.

**Figure 6 polymers-14-02491-f006:**
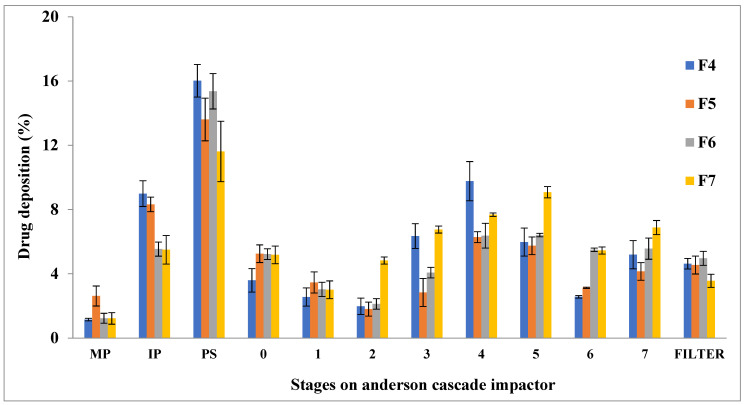
In vitro aerosol deposition pattern of microparticles at various stages of ACI.

**Table 1 polymers-14-02491-t001:** Formulation designs of spray-dried rifampicin-loaded microparticles.

Sample	Rifampicin (g)	Aloe Vera Powder (g)	Sodium Alginate (g)	L-Leucine (g)	Alginate: L-Leucine (*w*:*w*)
F1	0.20	0.02	0.05	0.05	1:1
F2	0.20	0.02	0.05	0.10	1:2
F3	0.20	0.02	0.05	0.15	1:3
F4	0.20	0.02	0.05	0.20	1:4
F5	0.20	0.02	0.05	0.25	1:5
F6	0.20	0.02	0.25	0.05	5:1
F7	0.20	0.02	0.20	0.05	4:1
F8	0.20	0.02	0.15	0.05	3:1
F9	0.20	0.02	0.10	0.05	2:1
Pure drug	0.20	NA	NA	NA	NA

**Table 2 polymers-14-02491-t002:** Particle size, PDI, zeta potential and percentage yield of microparticles.

Formulation Code	Particle Size (µm)	PDI	Zeta Potential (mV)	Roughness (nm)	Circularity	Yield (%)
F1	2.41 ± 0.79	0.04 ± 0.00	−0.78 ± 0.14	1.13 ± 0.40	0.68 ± 0.02	60.47 ± 5.01
F2	2.82 ± 0.92	0.03 ± 0.00	−0.61 ± 0.24	0.99 ± 0.39	0.74 ± 0.06	56.53 ± 4.03
F3	3.69 ± 0.53	0.02 ± 0.00	−0.33 ± 0.12	0.88 ± 0.38	0.77 ± 0.00	48.38 ± 5.06
F4	2.62 ± 0.62	0.05 ± 0.00	−0.86 ± 0.18	0.85 ± 0.12	0.74 ± 0.02	46.50 ± 8.01
F5	2.08 ± 0.71	0.03 ± 0.00	1.69 ± 0.11	1.20 ± 0.22	0.83 ± 0.01	59.51 ± 6.01
F6	3.21 ± 0.89	0.03 ± 0.00	−0.26 ± 0.21	1.14 ± 0.22	0.89 ± 0.01	65.47 ± 5.01
F7	2.82 ± 0.55	0.04 ± 0.00	−0.40 ± 0.09	0.77 ± 0.33	0.94 ± 0.01	60.76 ± 7.04
F8	2.93 ± 0.95	0.03 ± 0.00	−0.46 ± 0.18	0.76 ± 0.14	0.72 ± 0.02	55.11 ± 6.03
F9	2.65 ± 0.78	0.02 ± 0.00	−0.61 ± 0.148	0.67 ± 0.52	0.96 ± 0.03	52.07 ± 4.01
Pure drug	3.19 ± 0.53	0.04 ± 0.00	0.28 ± 0.22	1.44 ± 0.23	0.46 ± 0.02	45.87 ± 2.01

**Table 3 polymers-14-02491-t003:** Drug content, density and flow characteristics of the spray-dried microparticles.

Formulation Code	Drug Content (mg/mg)	Association Efficiency (%)	Bulk Density (g/mL)	Tapped Density (g/mL)	Carr’s Index (%)	Hausner’s Ratio
F1	0.245 ± 0.008	39.28 ± 1.064	0.20 ± 0.01	0.47 ± 0.03	57.50	2.35
F2	0.265 ± 0.010	49.08 ± 1.91	0.18 ± 0.03	0.50 ± 0.02	64.30	2.80
F3	0.373 ± 0.012	78.42 ± 2.57	0.13 ± 0.01	0.38 ± 0.03	65.00	2.86
F4	0.397 ± 0.011	93.36 ± 2.52	0.15 ± 0.01	0.37 ± 0.04	60.00	2.50
F5	0.368 ± 0.007	95.68 ± 1.94	0.25 ± 0.05	0.40 ± 0.03	37.50	1.60
F6	0.370 ± 0.008	96.15 ± 2.15	0.18 ± 0.04	0.35 ± 0.05	47.99	1.92
F7	0.373 ± 0.007	87.65 ± 1.59	0.12 ± 0.01	0.21 ± 0.03	42.85	1.75
F8	0.355 ± 0.008	74.64 ± 1.78	0.08 ± 0.01	0.20 ± 0.05	60.00	2.50
F9	0.332 ± 0.009	61.62 ± 1.77	0.27 ± 0.03	0.43 ± 0.02	37.21	1.61

**Table 4 polymers-14-02491-t004:** The correlation coefficient (R^2^), *n* value and rate constant (K) of Korsmeyer–Peppas model equation for spray-dried rifampicin microparticles.

Formulation	R^2^	*n*	K	Mechanism
F4	0.9756	0.6429	0.024 ± 0.0565	non-Fickian mechanism
F5	0.9792	0.7335	0.076 ± 0.1868	non-Fickian mechanism
F6	0.9972	0.8496	0.353 ± 0.8873	non-Fickian mechanism
F7	0.9789	0.6461	0.023 ± 0.0653	non-Fickian mechanism

**Table 5 polymers-14-02491-t005:** Aerodynamic performance of the spray-dried rifampicin loaded microparticles.

Aerodynamic Parameters	F4	F5	F6	F7
D_aer_	1.03	2.025	1.571	1.345
MMAD	1.44 ± 0.03	1.59 ± 0.05	1.42 ± 0.02	1.59 ± 0.03
GSD	3.16 ± 0.02	4.08 ± 0.07	3.79 ± 0.05	3.13 ± 0.07
TD (mg)	7.94	7.36	7.40	7.46
ED (mg)	5.46 ± 0.25	4.54 ± 0.28	4.84 ± 0.56	5.28 ± 0.14
DD (mg)	3.38 ± 0.18	2.74 ± 0.19	3.20 ± 0.21	3.91 ± 0.11
PD	68.72 ± 3.85	61.72 ± 4.34	65.35 ± 3.19	70.71± 5.39
PI	42.54 ± 2.25	37.18 ± 1.95	43.22 ± 2.23	52.39 ± 2.95
A. Mass deposited on all stages				
FPD	3.38 ± 0.16	2.74 ± 0.25	3.20 ± 0.21	3.91 ± 0.15
FPF	61.95 ± 4.78	60.24 ± 3.25	66.15 ± 2.25	74.09 ± 2.33
RF	100 ± 0.00	100	100	100
B. Mass deposited on stage 2 and below				
FPD	3.09 ± 0.15	2.35 ± 0.18	2.81 ± 0.19	3.52 ± 0.13
FPF	56.72 ± 4.03	51.74 ± 3.43	58.16 ± 2.93	66.77 ± 2.44
RF	91.57 ± 2.38	85.89 ± 3.03	87.93 ± 4.11	90.12 ± 3.93
C. Mass deposited on stage 2 and below				
FPD	2.73 ± 0.14	1.96 ± 0.24	2.43 ± 0.17	2.94 ± 0.14
FPF	50.13 ± 3.97	43.22 ± 4.07	50.28 ± 3.15	55.70 ± 2.17
RF	80.93 ± 3.98	71.74 ± 4.57	76.01 ± 2.27	75.18 ± 2.44
D. Aerosol mass in the range of ≥ 0.5 ≤ 3µm				
FPD	1.96 ± 0.13	1.32 ± 0.11	1.65 ± 0.19	2.16 ± 0.06
FPF	35.97 ± 0.16	29.14 ± 1.22	34.18 ± 1.34	40.94 ± 0.22
RF	57.90 ± 1.89	48.37 ± 2.42	51.68 ± 1.98	55.26 ± 1.03

## Data Availability

All data, models and code generated or used during the study appear in the published article.

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
