# Peer review of "Polymeric Microparticles: Synthesis, Characterization and In Vitro Evaluation for Pulmonary Delivery of Rifampicin"

_polymers, 2022, doi:10.3390/polym14122491_

Round 1

Reviewer 1 Report

This work was well designed, organized and clearly presented. It deals with the encapsulation of a Rifampicin drug in polymeric microparticles via spray drying and then studies the release behaviors and other properties. The results are interesting. I would like to recommend its publication in the journal after some minor revisions.

1)      The molecular structures of the drug and the carries should be schematically shown and their chemical interactions should be briefly discussed.

2)      What is the role of the amino acid in the drug-loaded microparticles? Control samples without the addition of the amino acid, as well as without the addition of Aloe vera powder, should be compared in terms of properties and release behaviors.

3)      Why the morphology for the different formulations is so different?

4)      During the spray drying process, would the three component be well mixed or phase-separated. That is, what about the spatial distribution of the three component in the sprat-dried microparticles.

5)      Equation (3) seems not correct.

Author Response

Reviewer 1 Comment Answer Location 1) The molecular structures of the drug and the carries should be schematically shown and their chemical interactions should be briefly discussed. Thank you for your comment. Schematic representation of drug and carrier as suggested are shown in figure 4. In section 3.8. FTIR analysis (yellow highlighted). 2) What is the role of the amino acid in the drug-loaded microparticles? Control samples without the addition of the amino acid, as well as without the addition of Aloe vera powder, should be compared in terms of properties and release behaviors. Thank you for your comment. 1. The hydrophobic L-leucine promotes the formation and integrity of microparticles prepared by the spray drying method due to the prevention of particle aggregation. Further, it promotes drug aerosolization from dry powder inhaler and has been previously used as an excipient for long-term storage of inhalation powders. 2. Due to the lower percentage yield of powder without addition of leucine in spray-dried formulations (results not included) in formulation, the ratio of sodium alginate and leucine was varied as mentioned in Table 1. 3. The aloe vera had been used as integral part of formulation due to its anti-microbial properties. 4. Reference added to elaborate role of amino acid in inhaled powder formulations 37. reference (Thiyagarajan, D., Huck, B., Nothdurft, B., Koch, M., Rudolph, D., Rutschmann, M., ... & Lehr, C. M. (2021). Spray-dried lactose-leucine microparticles for pulmonary delivery of antimycobacterial nanopharmaceuticals. Drug delivery and translational research, 11(4), 1766-1778.) Reason for using amino acid and aloe vera as integral part of formulations are yellow highlighted in introduction. Thank you for your comment. The morphology of spray-dried particles changed with increasing concentrations of Leucine (F1 to F5) or sodium alginate (F9 to F6). The circularity of particles increased while the agglomeration tendency of the powder decreased with increasing concentration of leucine. As the concentration of sodium alginate increased with respect to leucine in spray drying feed solution, the particles became porous inside, rounded with cup-shaped morphology. Thus at higher leucine and sodium alginate concentrations, the particles get amorphous and morphology changes. The effect of increasing the concentration of leucine and sodium alginate on the morphology of spray-dried particle morphing had been elaborated in section 3.3 of the results and discussion section (yellow highlighted). 4) During the spray drying process, would the three components be well mixed or phase-separated. That is, what about the spatial distribution of the three component in the spray-dried microparticles. Thank you for your comment. The spray-dried feed solution was mixed completely and spray-dried as homogeneous solution of the three components. The methodology of preparing spray-dried feed solution was elaborated in methodology section 2.2.2 (yellow highlighted). 5) Equation (3) seems not correct. Thank you for your comment. Corrected as suggested Methodology section 2.2.7 (yellow highlighted)

Reviewer 2 Report

The present manuscript describes the production of rifampicin microparticles for pulmonary delivery aiming to develop an alternative route of administration for the treatment of tuberculosis of this antibiotic.

Here are some comments:

11)   In lines 89-91 the sentence “The essential amino acids L-leucine has been widely used in the formulation of dry powder and are generally regarded as a safe excipient for pulmonary administration.” A reference should be included

2   2) In Table 1 : “gm” should be replaced by “g”, the SI unit symbol for gram. In the last column “Alginate: Leucine”, the ratio should be included. Is it (w:w) ?

3     3) In Table 3: “Association efficiency (%age)” replace by “Association efficiency (%)”

4  4) In methods section, 2.2.12. The in-vitro drug release profile  lines 260-262 : “The reagent bottle was filled with 300 ml of phosphate buffer, and precisely known weight of microparticles previously packed in already activated dialysis membrane (MWCO 12000) were added to the dissolution medium”  Please complete the information. In addition, which was the amount of rifampicin used in the free form.

5     5) In lines 543-545 the sentence: “In the case of formulations (F4-F5) with higher leucine concentration, the release was significantly retarded when compared to formulations with higher concentrations of sodium”. If statistically significant differences were observed please include that information in the respective figure.

6     6) The results presented in Figure 6 correspond to how many independent experiments?  Standard deviation is missing. The quality of the graph should be improved.

1      7) The English grammar should be revised in the manuscript. In the title “pulomary” please change to “pulmonary”.

Author Response

Reviewer 2 Comment Answer Location In lines 89-91 the sentence “The essential amino acids L-leucine has been widely used in the formulation of dry powder and are generally regarded as a safe excipient for pulmonary administration.” A reference should be included. Thank you for your comment. Included as suggested Reference 33. Alhajj, N., O'Reilly, N. J., & Cathcart, H. (2021). Leucine as an excipient in spray dried powder for inhalation. Drug Discovery Today, 26(10), 2384-2396 In Table 1 : “gm” should be replaced by “g”, the SI unit symbol for gram. In the last column “Alginate: Leucine”, the ratio should be included. Is it (w:w) ? Thank you for your comment. 1. Replaced “gm” by g” as suggested. 2. The ratio is w:w Table 1 in methodology In Table 3: “Association efficiency (%age)” replace by “Association efficiency (%)” Thank you for your comment. Replaced as suggested Table 3 in results and discussion In methods section, 2.2.12. The in-vitro drug release profile lines 260-262 : “The reagent bottle was filled with 300 ml of phosphate buffer, and precisely known weight of microparticles previously packed in already activated dialysis membrane (MWCO 12000) were added to the dissolution medium” Please complete the information. In addition, which was the amount of rifampicin used in the free form. Thank you for your comment. 1. The dialysis method is the most versatile and popular to assess drug release from nano- and micro-sized dosage forms In this method, physical separation of the dosage forms is achieved by usage of a dialysis membrane which allows for ease of sampling at periodic intervals. 2. Method elaborated as suggested 3. reference 52 D’Souza, S. (2014). A review of in vitro drug release test methods for nano-sized dosage forms. Advances in Pharmaceutics, 2014. 4. The amount of rifampicin used in release study was 50 mg. Elaborated in section 2.2.12 In lines 543-545 the sentence: “In the case of formulations (F4-F5) with higher leucine concentration, the release was significantly retarded when compared to formulations with higher concentrations of sodium”. If statistically significant differences were observed please include that information in the respective figure. Thank you for your comment. 1. The statistically significant difference was observed as depicted p ≤ 0.05. 2. Corrected as suggested The p value added in the results and discussion section 3.9. The results presented in Figure 6 correspond to how many independent experiments? Standard deviation is missing. The quality of the graph should be improved. Thank you for your comment. 1. The results are shown as mean of triplicate experiments 2. Standard deviation added as suggested. 3. The quality of graph improved as suggested. Table 6 in results and discussion section. 1. The English grammar should be revised in the manuscript. In the title “pulomary” please change to “pulmonary”. Thank you for your comment. 1. The English grammar is checked and corrected. 2. The title corrected as suggested. The title and the whole text of article were double checked and the corrections are highlighted.
